# PRUNING CNNS WITH GRAPH RANDOM WALK & RANDOM MATRIX THEORY

## ABSTRACT

To facilitate the deployment of convolutional neural networks on resource-limited devices, filter pruning has emerged as an effective strategy because of its enabled practical acceleration. Evaluating the importance of filters is a crucial challenge in this field. Most existing works on filter pruning assess the relationships of filters using pairwise measures such as Euclidean distance and cosine correlation, which may not capture the global information within the layer. In this paper, we propose a novel filter pruning method, which leverages a graph-based approach to model the relationships among filters in convolutional layers. Each filter is represented as a node in a directed graph, and the edges between nodes capture the linear dependencies between filters. This structure allows us to assess the relative importance of each filter by conducting a random walk on the graph. Filters that exhibit weaker connections to others are considered less important and are pruned with minimal impact on model performance. Furthermore, we examine the eigenvalue spectrum of the adjacency matrix and observe a distribution similar to that of the spiked models in random matrix theory. This suggests that the spiked eigenvalues could serve as a significant indicator of the importance of each convolutional layer. We conduct image classification on CIFAR-10 and ImageNet to demonstrate the superiority of our method over the state-of-the-arts.

## 1 INTRODUCTION

Deep Convolutional Neural Networks (DCNNs) have revolutionized the field of computer vision by achieving state-of-the-art performance in various tasks (Simonyan & Zisserman, 2014; He et al., 2015; Szegedy et al., 2014). However, deploying resource heavy CNNs on devices with limited computational and storage capacities presents significant challenges. Consequently, numerous studies have focused on model compression and CNN acceleration, mainly including network pruning (He et al., 2018b; Lin et al., 2020; Sui et al., 2021), model quantization (Liu et al., 2018; Qin et al., 2020; Liu et al., 2020), low-rank decomposition (Jaderberg et al., 2014; Lin et al., 2016) and knowledge distillation (Shen et al., 2019; Hinton et al., 2015). Among them, network pruning has been widely studied due to its easy implementation and effective results.

Recent developments on pruning can be divided into two categories, i.e., weight pruning and filter pruning. Weight pruning removes individual filter weights, creating a sparse network that often requires special hardware to achieve acceleration, as it cannot efficiently utilize standard BLAS libraries. In contrast, filter pruning, which compresses the model by directly removing selected filters, maintains regular structures and is widely used due to its ability to enhance acceleration on general-purpose hardware.

Generally, there are two essential issues in the filter pruning, i.e., the layer importance measurement and the filter importance measurement. For the first issue, layer importance measurement is related to the per-layer pruning rate. Existing works such as (Chin et al., 2018; Lin et al., 2022) utilize different weight-oriented strategies to evaluate the importance of each convolutional layer based on pre-trained models. For the second issue, the filter importance measurement identifies which filters in the pre-trained model should be preserved and inherited to initialize the pruned network structure. Previous works (Ye et al., 2018; He et al., 2018a) performs filter pruning by following the "smaller-norm-less-important" criterion, which believes that filters with smaller norms can be pruned safely due to their less importance. However, this criterion appears overly simplistic and may

not always hold true in all cases. Beyond this, He et al. (2018b) assumes the filters that are close to the geometric median are redundant, which is implemented by calculating the distance between filters pairs, Lin et al. (2022) introduces a recommendation-based filter selection scheme where each filter recommends a group of its closest filters, these two methods both try to find the central filters in certain measure spaces, which is proved to be effective by experiments. Joo et al. (2021) proposes Linearly Replaceable Filter (LRF), which suggests that a filter that can be approximated by the linear combination of other filters is replaceable, however they only focus on the approximated error and haven't dived into the relationships among filters.

More recently, graph-driven methods have been developed to identify important filters and achieve competitive performance. For example, Li et al. (2023) utilize a graphical model to represent the similarity relationships between the output feature maps of filters, while Shi et al. (2023) introduce Von Neumann graph entropy as a novel measure for filter importance. However, most of these approaches predominantly rely on similarity metrics like Euclidean distance or cosine similarity to measure the similarity between filters, which only considers the pairwise relationships between filters, overlooking the interactions among them within the whole layer.

To address the aforementioned limitations, we proposed a novel method to evaluate the importance of filters by focusing on the interactions between them. Previous works have shown that some filters can have similar functionality, our approach assumes that each filter potentially shares parts of its functionality with others. Similar to the approach in (Joo et al., 2021), we model these interactions by linearly reconstructing each filter using its peers, as shown in Equation 1. From this reconstruction, the coefficients indicate the strength of connection between a filter and its peers, while the residuals represent the unique functionality of itself. Proceeding with this approach, we construct a graph where each node represents a filter, and the edges between them are weighted by the coefficients from the linear combinations. This graph representation allows us to apply network analysis techniques to find the central nodes. Filters that act as central nodes in this graph will be considered as hubs of functionality, playing pivotal roles in the layer. Figure 1 illustrates the pipeline of our pruning method. Additionally, we analyze the eigenvalue spectrum of the adjacency matrix and observe a distribution resembling that of spiked models in random matrix theory. This similarity suggests that the spiked eigenvalues may serve as key indicators of the significance of each convolutional layer, as detailed in the pipeline shown in Figure 2. Further elaboration on this method is provided in Section 4.

**Contributions.** The main contributions of this paper can be summarized as follows:

- Leveraging insights from Random Matrix theory, we present a novel approach for quantifying the importance of different layers, thereby facilitating the determination of the optimal pruning rate for each layer.

- By modeling filter relationships through a graph structure, our approach utilizes network analysis techniques to reveal deeper insights into the network's architecture. This enables the identification of key filters that serve as central nodes, elucidating their roles in contributing to the network's effectiveness.

- Applying our proposed method to different model pruning tasks on CIFAR-10 and ImageNet datasets, extensive experiments demonstrate the effectiveness of our pruning strategy, which outperforms other similar algorithms by achieving higher compression rates while maintaining high accuracy.

## 2 RELATED WORK

This section examines seminal works in the field of Convolutional Neural Network (CNN) pruning methodologies, distinguishing between Filters-Importance-Supported (FIS) and Layer-Importance-Supported (LIS) approaches. Subsequently, we will provide a concise overview of Random Matrix Theory.

### 2.1 FILTERS-IMPORTANCE-SUPPORTED (FIS) PRUNING METHODS

Filters-Importance-Supported (FIS) pruning methods can be categorized into two primary streams: Data-driven Pruning Methods and Data-Independent Pruning Methods.

Data-driven pruning methods rely on training data to determine which filters to be pruned. ThiNet Luo et al. (2017) adopts the statistics information from the next layer to guide the filter selections. Liu et al. (2017) introduces sparsity in the scaling factors of batch normalization layers to identify and prune minor significant channels during training. HRank Lin et al. (2020) states that weights corresponding to feature maps with high rank contain more important information and therefore need to be retained in the pruning process. Beyond this CHIP Sui et al. (2021) explores the importance of filters using intra-channel information and FPEI Wang et al. (2021) defines filter importance based on the entropy of the corresponding feature maps. By comparison, Data-Independent pruning approaches often leverage structural characteristics of the network or predefined rules. Li et al. (2017) utilizes $l_1$-norm criterion to prune unimportant filters. He et al. (2018a) proposes to select filters with a $l_2$-norm criterion and prune those selected filters in a soft manner. He et al. (2018b) evaluates the redundancy of filters by measuring their distance from the geometric median of the filter group within each layer. Lin et al. (2022) proposes Cross-Layer Ranking and k-Reciprocal Nearest Filter Selection to decide the pruning rate of each layer and the most important filters, resectively.

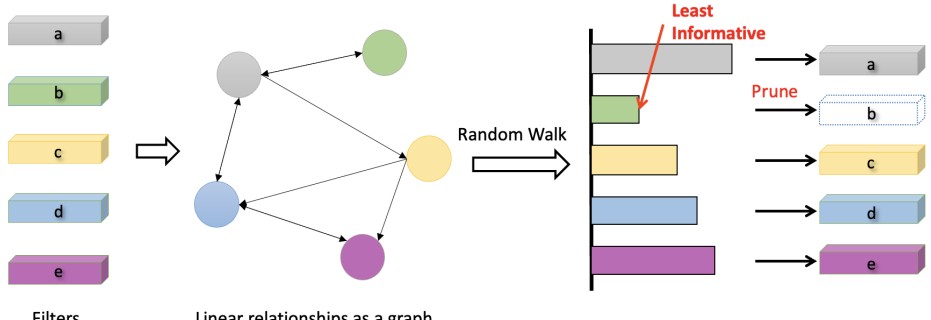

Figure 1: Filter importance measurement pipeline

## 2.2 LAYER-IMPORTANCE-SUPPORTED (LIS) PRUNING METHODS

Layer Importance Supported (LIS) methods focus on evaluating the significance of individual layers before proceeding to filter removal based on these assessments. For instance, studies such as (Li et al., 2017; Mao et al., 2017), and (Suau et al., 2020) employ various weight-oriented strategies to gauge the importance of convolutional layers in fully-trained models. The research by Li et al. (2017) investigates the sensitivity of each convolutional layer through multiple layer-wise pruning experiments utilizing the $l_1$-norm, highlighting the critical role of sensitive layers while advocating for the removal of filters from less sensitive ones. In a similar vein, Mao et al. (2017) introduces a coarse-grained pruning methodology that facilitates iterative layer pruning based on sensitivity analysis. Additionally, Suau et al. (2020) applies principal filter analysis to derive a more compact model by leveraging the intrinsic correlations among filter responses within the layers.

## 2.3 RANDOM MATRIX THEORY

In the field of Random Matrix Theory (RMT), the spike model serves as a crucial framework for analyzing the separation of signal and noise in high-dimensional datasets. The spike model introduces a finite number of large eigenvalues, or "spikes," into the spectrum of a random matrix to represent significant underlying signals amidst random noise. Key studies in this area have focused on the distribution of these spike eigenvalues and their influence on the overall eigenvalue distribution of the matrix.

Baik et al. (2005) examined the asymptotic distribution of the largest eigenvalues in non-null complex sample covariance matrices, uncovering the phase transition phenomenon now known as the Baik–Ben Arous–Péché (BBP) transition. This work illustrates how spikes can lead to distinct eigenvalue behaviors compared to the bulk of the spectrum. Johnstone (2001) applied the spike model to high-dimensional statistics, particularly in the context of Principal Component Analysis (PCA). His analysis provided conditions under which spike eigenvalues can be reliably separated from noise eigenvalues, enhancing the effectiveness of PCA in signal detection. More recently, Benaych-Georges

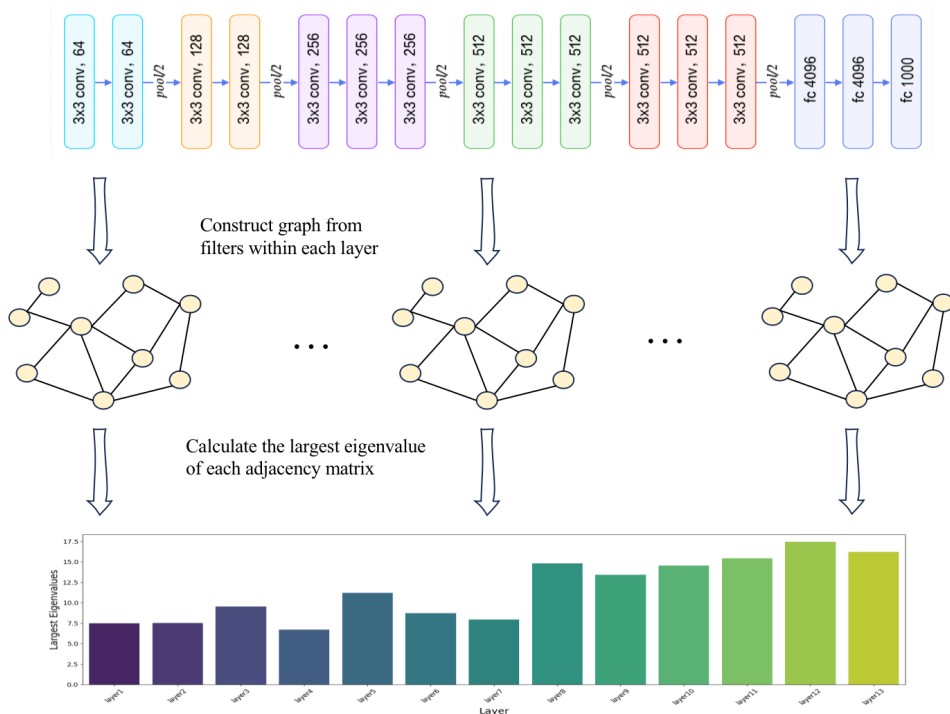

Figure 2: Layer importance measurement pipeline

& Nadakuditi (2011) extended the spike model to accommodate multiple spikes, exploring their interactions and the resulting modifications to the eigenvalue distribution. Their contributions have deepened the understanding of how multiple significant signals can coexist and influence the spectral properties of random matrices.

# 3   PRELIMINARY

This subsection introduces the symbols and notations used throughout our discussion. We consider a pre-trained Convolutional Neural Network (CNN) with $l$ convolutional layers, denoted as $L = \{L_1, L_2, \ldots, L_l\}$. Each layer $L_i$ contains $n_i$ filters: $L_i = \{f_i^1, f_i^2, \ldots, f_i^{n_i}\}$. Each filter $f_i^j$ is represented as a 3D tensor $f_i^j \in \mathbb{R}^{c_i \times h \times w}$, where $c_i$ is the number of channels (filters) from the previous layer, and $h$ and $w$ are the height and width of the filter respectively, which are constants. Since our analysis is conducted on each convolutional layer independently, for simplicity, we will omit the convolutional layer index $i$ henceforth and refer to the $j$-th filter in the layer simply as $f_j$ ($j = 1 \ldots n$). Finally, we flatten each filter to a vector : $f_j \in \mathbb{R}^{c \cdot h \cdot w}$.

## 3.1   MARCHENKO-PASTUR LAW AND SPIKED COVARIANCE MODEL

The Marchenko-Pastur law is a key result in random matrix theory, describing the limiting distribution of eigenvalues for large sample covariance matrices. For a large random matrix $\mathbf{X} \in \mathbb{R}^{p \times n}$, where the entries are independent and identically distributed (i.i.d.) with zero mean and unit variance, the eigenvalues of $\frac{1}{n}\mathbf{X}\mathbf{X}^\top$ follow a limiting distribution as both $n, p \to \infty$ with the ratio $p/n \to c \in (0, \infty)$. This distribution, known as the Marchenko-Pastur distribution, has a density function:

$$\rho_c(\lambda) = \frac{\sqrt{(\lambda - \lambda_-)(\lambda_+ - \lambda)}}{2\pi c\lambda}, \quad \lambda \in [\lambda_-, \lambda_+],$$

where $\lambda_- = (1 - \sqrt{c})^2$ and $\lambda_+ = (1 + \sqrt{c})^2$. The eigenvalues are supported in $[(1 - \sqrt{c})^2, (1 + \sqrt{c})^2]$. For $c < 1$, some eigenvalues are exactly zero, with a probability mass at 0.

In the spiked covariance model, certain population eigenvalues $\ell_i$ of the covariance matrix $\mathbf{C}$ deviate from the bulk of the Marchenko-Pastur distribution. These "spikes" correspond to strong signals, causing deviations in the sample eigenvalues. According to Theorem 2.13 in Couillet & Liao (2022), if $\ell_i > \sqrt{c}$, the sample eigenvalue $\hat{\lambda}_i$ converges to:

$$\hat{\lambda}_i \xrightarrow{a.s.} 1 + \ell_i + c\frac{1+\ell_i}{\ell_i}.$$

If $\ell_i \leq \sqrt{c}$, the sample eigenvalue stays within the Marchenko-Pastur bulk, behaving asymptotically as $(1 + \sqrt{c})^2$ which will be contained in the bulk. Figue 3 shows a example of eigenvalue distribution of Spiked model.

From a noise-signal perspective, the Marchenko-Pastur law helps distinguish between noise (represented by the bulk of the distribution) and meaningful signals (represented by the spiked eigenvalues). Eigenvalues within the bulk correspond to noise, while those exceeding $\sqrt{c}$ indicate strong signals separating from the noise.

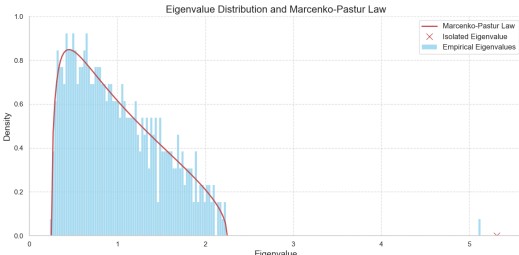

Figure 3: eigenvalue distribution of Spiked model

## 4 METHOD

### 4.1 GRAPH CONSTRUCTION BASED ON LINEAR DEPENDENCIES AMONG FILTERS

To accurately model the relationships among filters within the CNN, we examine the linear relationships between filters in each layer. Each filter $f_j$ can be approximated by a linear combination of other filters in the same layer:

$$f_j = \sum_{l \neq j} \alpha_{j,l} f_l + \varepsilon_j, \tag{1}$$

Based on linear combination, we establish a directed weighted graph denoted by $\mathcal{G}$. In this graph, each node represents a filter, and the directed edges between nodes are weighted according to the absolute values of the coefficients $\alpha_{j,l}$, which indicate the strength of connection between the filters. The weight $W_{j,l}$ of a directed edge from filter $j$ to filter $l$ is defined as follows:

$$w_{j,l} = |\alpha_{j,l}| \tag{2}$$

This representation allows us to capture the mutual relationships among filters from a graph perspective.

### 4.2 EVALUATING LAYER IMPORTANCE VIA EIGENVALUE ANALYSIS (INSPIRED BY RMT)

To quantify the importance of each layer, we perform eigenvalue analysis on the weighted adjacency matrix $\mathbf{W}$ associated with the connections between filters. The eigenvalues of this matrix provide insights into the structure of the network. Formally, we compute the eigenvalues $\lambda_1, \lambda_2, \ldots, \lambda_n$ by solving the following equation:

$$\mathbf{W}\mathbf{v}_i = \lambda_i \mathbf{v}_i, \tag{3}$$

where $\lambda_i$ represents the $i$-th eigenvalue and $\mathbf{v}_i$ is its corresponding eigenvector. In particular, the largest eigenvalue $\lambda_{\max}$ plays a significant role in capturing the dominant structural properties of the matrix.

During our experiments, we observed that the distribution of eigenvalues exhibits characteristics similar to the spiked covariance model from Random Matrix Theory (RMT). This resemblance suggests that we can draw a distinction between the signal-dominated components and the noise-dominated components of the matrix. In the spiked model, the spiked eigenvalues correspond to significant "signal" components, while the remaining eigenvalues, clustered around a bulk distribution, represent "noise" fluctuations. Thus, the largest eigenvalue $\lambda_{\max}$ offers a meaningful way to quantify the importance of the layer by highlighting the amount of structured, non-random information retained within the connections. We define Layer Importance Score(LIS) for each layer:

$$LIS = \lambda_{max} \qquad (4)$$

Further more, we transform LIS into prunning rate for each layer

$$PR_l = 1 - (LIS_l/LIS_1)/N_l * \gamma \qquad (5)$$

$PR_l$ and $N_l$ is the pruning rate and the filters number for $l$-th layer respectively, $\gamma$ is a parameter to control the pruning strength. We will present an example of the pruning rate for VGG-16 in table 1.

---

**Algorithm 1** Layer importance measurement via eigenvalue analysis

---

1: **Input:** A pre-trained CNN with $L$ layers.
2: **Output:** the pruning ratio of each layer $PR_l$.
3: Initialize an empty list $PR$ to keep track of pruning ratio of each layer.
4: **for** $i = 1$ to $L$ **do**
5:     Compute the weighted adjacency matrix $W_i$ based on Equation 2.
6:     Compute the eigenvalues and eigenvectors of $W_i$ based on Equation 6.
7:     Define the layer importance score as the biggest eigenvalue $LIS = \lambda_{max}$.
8: **end for**
9: Compute the pruning ratio $PR$ for each layer based on Equation 8.
10: **return** Pruning ratio list $PR$.

---

## 4.3 Evaluating Filter Importance via Random Walks (Inspired by PageRank)

In graph theory, a node's importance is often inferred from its connectivity. Drawing inspiration from the PageRank algorithm, we propose a random walk framework to evaluate the significance of filters (nodes) within the network. Random walk is a fundamental concept where a "walker" starts at a given node and randomly transitions to adjacent nodes based on predefined probabilities. The frequency with which the walker visits each node reflects its importance in the graph. In our framework, edge weights are used to represent the strength of connections between filters, which in turn dictate the transition probabilities.

More formally, in a random walk, the probability of transitioning from filter $i$ to filter $j$ is proportional to the edge weight between them. The random walker thus traverses the network, spending more time on filters that are better connected or more influential, thereby capturing their relative significance. Over many iterations, the stationary distribution of the walker's visits can be interpreted as a measure of each filter's importance, similar to how PageRank ranks webpages based on link structure. Based on the previous discussion, we define a transition matrix $\mathbf{P}$, where each element $p_{ij}$ represents the probability of moving from filter $i$ to filter $j$. The self-transition probability $p_{jj}$, based on the $L_2$-norm of the residual for filter $j$, captures the filter's retention of its unique influence:

$$p_{jj} = \|\epsilon_j\|_2 \qquad (6)$$

For transitions between different filters $j$ and $l$, the probability is proportional to the edge weight $W_{jl}$, ensuring the total transition probabilities sum to one:

$$p_{jl} = \frac{w_{jl}}{\sum_{k \neq j} w_{jk}} \cdot (1 - \|\epsilon_j\|_2) \tag{7}$$

We simulate the random walk by iteratively applying the matrix $\mathbf{P}$ to an initial probability distribution $\mathbf{v}^{(t)}$, where $\mathbf{v}^{(t+1)} = \mathbf{P}\mathbf{v}^{(t)}$, until it converges to the stationary distribution $\mathbf{v}^*$. The stationary distribution reflects the long-term behavior of the walk, revealing the most visited (i.e., influential) filters in the network:

$$\mathbf{v}^* = \mathbf{P}\mathbf{v}^* \tag{8}$$

Filters with higher probabilities in $\mathbf{v}^*$ act as key nodes or hubs, providing insights into which filters are most crucial. These high-ranking filters can then be prioritized, while those with lower significance will be pruned with minimal impact on network performance.

---

**Algorithm 2** Graph-based filter pruning algorithm for CNN Compression

---

1: **Input:** A pre-trained CNN with $L$ layers; the pruning rate for each layer $PR_l$
2: **Output:** A pruned network with information-critical filters retained.
3: Initialize an empty list $kept\_filters$ to keep track of filters to retain in each layer.
4: **for** $i = 1$ to $L$ **do**
5:     **for** $j = 1$ to $n_i$ **do**
6:         Compute the linear representation coefficients $\lambda_{j,l}$ and residuals $\varepsilon_j$ via Equation 2.
7:     **end for**
8:     Construct weighted graph based on Equation 1 and Equation 2.
9:     Construct the Random Walk transition matrix $P$ via Equation 3 and 4.
10:     Compute the stable distribution $v^*$ via Equation 5.
11:     Select the top $c_i$ filters based on $v^*$ and append to $kept\_filters$.
12: **end for**
13: Prune the network by removing filters not in $kept\_filters$.
14: Optionally fine-tune the network to recover performance.
15: **return** Pruned network

---

## 5 EXPERIMENTS

### 5.1 EXPERIMENTAL SETTINGS

**Baselines Models and Datasets.** To demonstrate the effectiveness and generality of our proposed channel independence-based approach, we evaluate its pruning performance for various baseline models on different image classification datasets. To be specific, we conduct experiments with three CNN models (ResNet-56, ResNet-110 and VGG-16) on CIFAR-10 dataset. What's more, we further evaluate our approach and compare its performance with other state-of-the-art pruning methods with ResNet-50 model on large-scale ImageNet dataset Deng et al. (2009).

**Pruning and Fine-tuning Configurations.** We conduct our empirical evaluations on Nvidia RTX A6000 GPUs with PyTorch 1.13 framework. After performing the channel graph-based filter pruning, we then perform fine-tuning on the pruned models with Stochastic Gradient Descent (SGD) as the optimizer. To be specific, we perform the fine-tuning for 400 epochs on CIFAR-10 datasets with the batch size, momentum, weight decay and initial learning rate as 128, 0.9, 0.05 and 0.01, respectively. On the ImageNet dataset, fine-tuning is performed for 180 epochs with the batch size, momentum, weight decay and initial learning rate as 256, 0.99, 0.0001 and 0.1, respectively.

### 5.2 EIGENVALUE DISTRIBUTION FOR LAYERS OF VGG-16

Figure 4 illustrates the eigenvalue distribution across each convolutional layer of VGG-16, which closely parallels the distribution observed in the spiked model depicted in Figure 3. Notably, each layer exhibits a pronounced spiked eigenvalue, indicative of the underlying signal's strength. Figure 5 presents the largest eigenvalue alongside the number of filters for each layer of VGG-16.

By varying the pruning strength parameter $\gamma$ in equation 5, we can achieve distinct pruning rates for each layer. Below are several illustrative examples:

| Layer | 1 | 2 | 3 | 4 | 5 | 6 | 7 | 8 | 9 | 10 | 11 | 12 | 13 |
|---|---|---|---|---|---|---|---|---|---|---|---|---|---|
| $\gamma = 64$ | 0.0 | 0.0 | 0.36 | 0.55 | 0.62 | 0.71 | 0.73 | 0.75 | 0.78 | 0.76 | 0.74 | 0.71 | 0.73 |
| $\gamma = 60$ | 0.06 | 0.05 | 0.4 | 0.58 | 0.65 | 0.73 | 0.75 | 0.77 | 0.79 | 0.77 | 0.76 | 0.73 | 0.75 |
| $\gamma = 50$ | 0.22 | 0.21 | 0.5 | 0.65 | 0.71 | 0.77 | 0.79 | 0.81 | 0.82 | 0.81 | 0.8 | 0.77 | 0.79 |

Table 1: Comparison of $\gamma$ for different pruning rates in VGG-16

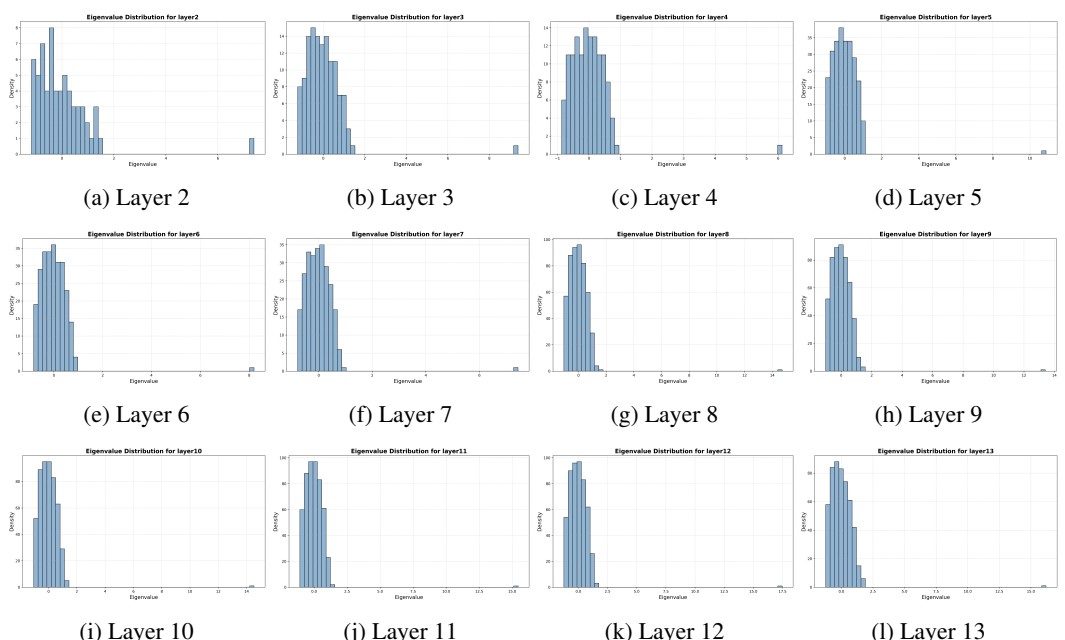

(a) Layer 2      (b) Layer 3      (c) Layer 4      (d) Layer 5

(e) Layer 6      (f) Layer 7      (g) Layer 8      (h) Layer 9

(i) Layer 10      (j) Layer 11      (k) Layer 12      (l) Layer 13

Figure 4: Eigenvalue distribution across different layers in VGG-16.

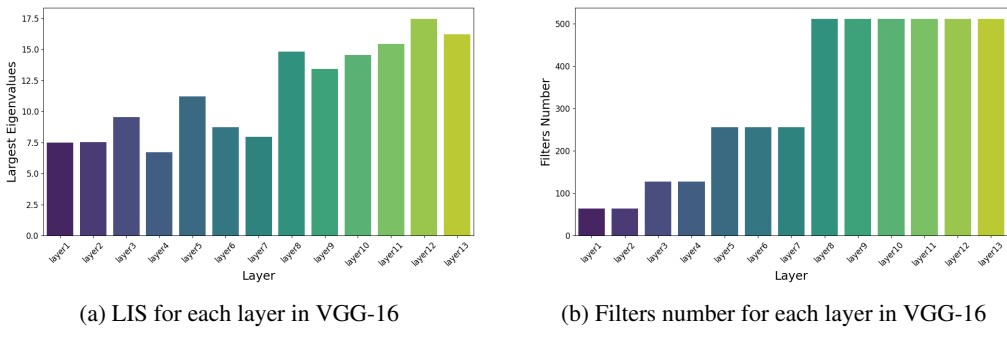

(a) LIS for each layer in VGG-16      (b) Filters number for each layer in VGG-16

Figure 5: LIS and filters number across different layers in VGG-16.

## 5.3 EVALUATION ON CIFAR-10 DATASET

Table 2 presents the experimental results on the CIFAR-10 dataset. The values in parentheses represent the reduction ratios. For some experiments, we adopted the same pruning rate as in previous works. The experiments marked with $\gamma$ indicate that the pruning rate was set based on our layer-importance quantification method.

For **VGG-16** model, our method achieves 0.19% increase in accuracy over the baseline model with 81.6% and 58.1% model parameters and FLOPs reductions, respectively. With the same reduction

Table 2: Experimental results on CIFAR-10 dataset.

| Method | Top-1 Accuracy (%) | | | Params (↓%) | FLOPs (↓%) |
|---|---|---|---|---|---|
| | Baseline | Pruned | Δ | | |
| VGG-16 | | | | | |
| HRank Lin et al. (2020) | 93.96 | 93.43 | -0.53 | 2.51M (82.9) | 145.61M (53.5) |
| CHIP Sui et al. (2021) | 93.96 | 93.86 | -0.10 | 2.76M (81.6) | 131.17M (58.1) |
| **(Ours)** | 93.96 | **94.15** | **+0.19** | 2.76M (81.6) | 131.17M (58.1) |
| HRank Lin et al. (2020) | 93.96 | 91.23 | -2.73 | 1.78M (92.0) | 73.70M (76.5) |
| CHIP Sui et al. (2021) | 93.96 | 93.18 | -0.78 | 1.90M (87.3) | 66.95M (78.6) |
| **(Ours)** | 93.96 | **93.73** | **-0.23** | 1.90M (87.3) | 66.95M (78.6) |
| **(Ours** $\gamma = 50$**)** | 93.96 | **92.79** | **-1.17** | **0.75M (95.0)** | **49.38M (84.3)** |
| ResNet-56 | | | | | |
| HRank Lin et al. (2020) | 93.26 | 93.17 | -0.11 | 0.49M (42.4) | 62.72M (50.0) |
| CLR-RNF Lin et al. (2022) | 93.26 | 93.27 | +0.01 | 0.38M (55.5) | 54.00M (57.3) |
| **(Ours)** | 93.26 | **93.78** | **+0.52** | 0.38M (55.5) | 54.00M (57.3) |
| HRank Lin et al. (2020) | 93.26 | 90.72 | -2.54 | 0.27M (68.1) | 32.52M (74.1) |
| CHIP Sui et al. (2021) | 93.26 | 92.05 | -1.21 | 0.24M (71.8) | 34.79M (72.3) |
| **(Ours)** | 93.26 | **92.62** | **-0.64** | 0.24M (71.8) | 34.79M (72.3) |
| **(Ours** $\gamma = 13$**)** | 93.26 | **92.32** | **-0.94** | **0.21M(75.5)** | **29.51M (76.5)** |
| ResNet-110 | | | | | |
| GAL (2019) | 93.50 | 92.74 | -0.76 | 0.95M (44.8) | 130.20M (48.5) |
| HRank Lin et al. (2020) | 93.50 | 92.65 | -0.85 | 0.53M (68.7) | 79.30M (68.6) |
| CLR-RNF Lin et al. (2022) | 93.50 | 93.71 | +0.21 | 0.53M (69.1) | 86.80M (66.0) |
| **(Ours)** | 93.50 | **94.39** | **+0.83** | 0.53M (69.1) | 86.80M (66.0) |
| **(Ours** $\gamma = 12$**)** | 93.50 | **92.89** | **-0.61** | **0.35M (80.0)** | **46.07M (81.78)** |

Table 3: Experimental results on ImageNet dataset.

| Method | Top-1 Accuracy (%) | | | Top-5 Accuracy (%) | | | Params (↓%) | FLOPs (↓%) |
|---|---|---|---|---|---|---|---|---|
| | Baseline | Pruned | Δ | Baseline | Pruned | Δ | | |
| ThiNet | 72.88 | 72.04 | -0.84 | 91.14 | 90.67 | -0.47 | 33.7 | 36.8 |
| SFP | 76.15 | 74.61 | -1.54 | 92.87 | 92.06 | -0.81 | N/A | 41.8 |
| FPGM | 76.15 | 75.59 | -0.56 | 92.87 | 92.63 | -0.24 | 37.5 | 42.2 |
| C-SGD | 75.33 | 74.93 | -0.40 | 92.56 | 92.27 | -0.29 | N/A | 46.2 |
| GAL | 76.15 | 71.95 | -4.20 | 92.87 | 90.94 | -1.93 | 16.9 | 43 |
| RRBP | 76.10 | 73.00 | -3.10 | 92.90 | 91.00 | -1.90 | N/A | 54.5 |
| PFP | 76.13 | 75.91 | -0.22 | 92.87 | 92.81 | -0.06 | 18.1 | 10.8 |
| HRank | 76.15 | 74.98 | -1.17 | 92.87 | 92.33 | -0.54 | 36.6 | 43.7 |
| SCOP | 76.15 | 75.95 | -0.20 | 92.87 | 92.79 | -0.08 | 42.8 | 45.3 |
| CHIP | 76.15 | 76.30 | +0.15 | 92.87 | 93.02 | +0.15 | 40.8 | 44.8 |
| **(Ours)** | 76.15 | **76.38** | **+0.23** | 92.87 | **93.22** | **+0.35** | 40.8 | 44.8 |
| **(Ours)** | 76.15 | **75.43** | **-0.72** | 92.87 | **92.93** | **+0.06** | **56.7** | **62.8** |

rate, our method demonstrates superior performance compared to CHIP. Additionally, we explore more extreme pruning strengths by setting the parameter $\gamma$, which result in only a slight drop in accuracy.

For **ResNet-56** model, our method obtains an increase of 0.52% accuracy over the baseline with 55.5% and 57/3% model parameters and FLOPs reductions. This is higher than in the increase of accuracy of CLR-RNF of 0.01%. Moreover, even with higher model parameters and FLOPs reductions of 71.8% and 72.3%, it achieves 0.57% higher accuracy than CHIP with the same reduction rate.

For **ResNet-110**, our achieves an increase of 0.83% accuracy over the baseline with 69.1% and 66% model parameters and FLOPs reductions. With higher model parameters and FLOPs reductions of 80.0% and 81.78%, our model still achieves a minor decrease in accuracy of only 0.61%.

## 5.4 EVALUATION AND COMPARISON ON IMAGENET DATASET

Table 3 summarizes the pruning performance of our approach for ResNet-50 on the ImageNet dataset. Our method, targeting a moderate compression ratio, achieves a reduction of 40.8% in storage and 44.8% in computation, while simultaneously increasing the accuracy by 0.23% compared to the baseline model. Moreover, as we increase the compression ratio further, our approach continues to outperform state-of-the-art methods. For instance, compared to CHIP, our method achieves a higher accuracy increment of 0.08% in the moderate compression range and maintains the same level of accuracy in the high compression range, all while necessitating a significantly smaller model size and fewer FLOPs.

## 6 CONCLUSION

In this study, we introduce an innovative filter pruning method by integrating a graph-based method with insights from Random Matrix Theory (RMT). By modeling the relationships among filters as a graph and assessing filter importance through random walks, alongside evaluating layer significance via eigenvalue analysis of the adjacency matrix, our method offers a nuanced understanding of structural redundancies in CNNs. Extensive evaluations across various datasets demonstrate that our proposed approach not only quantifies filter and layer importance effectively but also brings significant reductions in storage and computational costs, all while preserving high model accuracy..

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
