# OpenReview forum: "PRUNING CNNS WITH GRAPH RANDOM WALK & RANDOM MATRIX THEORY"
_ICLR.cc/2025/Conference — ICLR 2025 Conference Withdrawn Submission_

### Official Review · Reviewer_rHJR · 2024-10-31

**Soundness:** 2
**Presentation:** 3
**Contribution:** 3
**Rating:** 3
**Confidence:** 4

**Summary:**

Traditional approaches often use pairwise measures to assess filter importance, which may overlook global relationships. This paper presents a graph-based method for filter pruning, representing each filter as a node in a directed graph and capturing linear dependencies through edges. Importance is evaluated via random walks on the graph. The method demonstrates superior performance in image classification tasks on CIFAR-10 and ImageNet compared to existing state-of-the-art techniques.

**Strengths:**

- The idea of modeling filter relationships through a graph structure is quite new.
- Assessing the relative importance of each filter by conducting a random walk on the graph is interesting.

**Weaknesses:**

- Wrong expression. In addition to weight pruning and filter pruning, there is also layer pruning.
- The baselines used for comparison are relatively old, mostly from 2022 and before.
- The title of Table 1 is below the table, while the title of Table 2 is above the table.
- There are only two tables on page 9, leaving a lot of blank space, making the article look rough.
- The model structure used for verification is too simple. Experiments were only conducted on ResNet-56, ResNet-110 and VGG-16. Can this filter pruning method be applied to existing large language models?
- What is the time cost of graph-based filter pruning algorithm?
- Does the number of filters affect the efficiency of the algorithm?
If the author can solve my above problems, I will be happy to increase the score.

**Questions:**

See the Weaknesses

---

### Official Review · Reviewer_hrMX · 2024-11-02

**Soundness:** 3
**Presentation:** 3
**Contribution:** 2
**Rating:** 5
**Confidence:** 4

**Summary:**

## Summary
The paper proposes a novel filter pruning method for Deep Learning CNNs that utilizes a graph-based approach combined with Random Matrix Theory (RMT) for effective pruning in the resource-limited condition. The method builds a graph where each filter is a node, and the connections between nodes represent linear dependencies among filters in the same layer, which is further analyzed using random walk techniques inspired by the PageRank algorithm. The experimental results on CIFAR-10 and ImageNet datasets show superior performance compared to other methods, achieving higher compression rates while maintaining a competitive accuracy.

**Strengths:**

## Strengths and Weaknesses

Pros:

1. The paper introduces a novel pruning strategy that integrates graph-based modeling with RMT in the field of filter pruning.

2. The system figures in figure 1 and figure 2 are easy to understand and helpful to present the implemented method in the paper.

3. Multiple evaluation benchmarks are considered to support the experimental results in the paper.

**Weaknesses:**

## Cons:

1. The description of the experimental results lacks clarity. For instance, in Table 2, how to select certain rows in bold is not explained. It is unclear why, for example, CHIP achieves better pruned accuracy in row 5 compared to the proposed method shown in row 7.

2. The experimental results do not include comparisons with more recent publications or other graph-based pruning methods, such as VNGEP.

3. The proposed method lacks a comprehensive theoretical analysis.

**Questions:**

## Questions for the Authors


Please also refer the comments in the strengths and weaknesses sections:

1. Can the proposed method be extended to transformer-based models, such as Vision Transformers (ViT)? If so, could the authors provide empirical results to demonstrate this?

2. Is the proposed method compatible with other optimizers beyond SGD? If so, which optimizers have been tested, and what was their impact on performance?

---

### Official Review · Reviewer_eccB · 2024-11-02

**Soundness:** 3
**Presentation:** 2
**Contribution:** 2
**Rating:** 5
**Confidence:** 3

**Summary:**

This paper proposes a CNN pruning method based on graph structure and random matrix theory, which enables efficient pruning by treating filters as nodes in a graph and using random wandering and eigenvalue analysis to determine the importance of filters and layers. This approach captures global information within the convolutional layers while reducing parameters and computation while maintaining model accuracy.

**Strengths:**

1. Efficient pruning with random walks. Leveraging a random walk framework similar to PageRank enables an intuitive and effective way to rank filters.
2. Model filter relationships by graph structure. It enables the identification of key filters that serve as central nodes.
3. The performances of the proposed method are significant.
4. The pictures in this paper are beautiful.

**Weaknesses:**

1. Limited datasets of experiments. Although the paper performs experiments on the CIFAR-10 and ImageNet datasets, it lacks validation on other types of datasets. This may limit the generality of the approach.
2. Comparison algorithms are few and not new enough.
3. The paper lacks an analysis of the algorithmic time complexity of the pruning process

**Questions:**

How do you correctly determine the importance of an edge? According to equation 2, if alpha is negative, edge weights is likely to be higher.

---

### Official Review · Reviewer_6HrX · 2024-11-02

**Soundness:** 2
**Presentation:** 2
**Contribution:** 2
**Rating:** 3
**Confidence:** 3

**Summary:**

This paper presents a filter pruning method for Convolutional Neural Networks (CNNs). Taditional approaches  primarily use pairwise similarity measures to determine filter importance. Unlike those, this method leverages a graph-based framework to model relationships among filters within a CNN layer. Each filter is represented as a node in a directed graph, with edges reflecting the linear dependencies between filters. The authors introduce two main contributions:

1. Graph-Based Filter Importance Measurement: By constructing a directed graph of filters, the authors apply a random walk (inspired by PageRank) to evaluate the significance of each filter. Filters with weaker connections, indicated by low visitation probabilities during the random walk, are deemed less important and pruned.

2. Layer Importance via Eigenvalue Spectrum: Using concepts from Random Matrix Theory  the paper proposes analyzing the eigenvalue spectrum of the graph's adjacency matrix. The largest eigenvalues, or spiked eigenvalues, serve as indicators of each layer's importance, guiding the pruning rate for each layer.

The authors validate through experiments on CIFAR-10 and ImageNet datasets using models like VGG-16 and ResNet. The results demonstrate that this approach achieves higher compression rates while maintaining or improving accuracy compared to state-of-the-art pruning techniques.

**Strengths:**

1. The paper presents a novel approach to filter pruning by integrating graph-based techniques with random matrix theory for evaluating filter and layer importance. Conventional pairwise similarity measures, such as Euclidean distance or cosine similarity are commonly used in filter pruning. By constructing a **filter graph** and leveraging **random walk** techniques inspired by PageRank, the method introduces a fresh perspective to model filter interactions within a CNN.
2. The methodology is well-defined. The experimental evaluation is based on multiple CNN architectures (e.g., VGG-16, ResNet) and datasets (CIFAR-10, ImageNet). The approach demonstrates slight performance improvements.
3. Overall, the paper is well-structured and provides a clear presentation of complex topics, making it accessible to readers with varied backgrounds in machine learning and graph theory.
4. The proposed approach could inspire new pruning strategies that explore other advanced graph metrics or probabilistic frameworks.

**Weaknesses:**

1. Limited Evaluation Across Diverse Architectures: While the method is tested on VGG-16 and ResNet models, both of which have relatively regular structures, the paper lacks experiments on more diverse architectures. Further, I find the performance improvements  not substantial.
2. Sensitivity to Hyperparameters: The paper uses fixed values for key hyperparameters, such as the random walk transition probabilities and the pruning strength parameter, without exploring their influence on model performance or robustness. Hyperparameters may play a significant role in its effectiveness, especially in controlling the extent and impact of pruning.Conducting a sensitivity analysis or ablation study on important hyperparameters, such as the graph edge weights or the random walk probabilities, could provide insights into how these settings impact performance across different tasks.
3. Scalability and Applicability to Larger Networks: The method is primarily evaluated on relatively small datasets like CIFAR-10 and moderately large networks like ResNet-50 on ImageNet. However, it remains unclear if the approach scales effectively to larger architectures (e.g., ResNet-152, EfficientNet) or more complex datasets. As model and dataset sizes increase, the computational overhead of constructing and analyzing a filter graph could become a bottleneck, potentially limiting scalability. To strengthen this analysis, I recommend that the authors provide a computational complexity analysis or empirical runtime comparisons with existing methods, especially for larger architectures. Additionally, evaluating the method on more datasets like CIFAR-100 or MS COCO, would provide a more comprehensive understanding of the approach’s scalability and applicability in real-world scenarios.
4. Limited Comparison with Other Graph-Based Pruning Methods: Although the paper briefly mentions recent graph-based pruning methods, such as the use of Von Neumann graph entropy (Shi et al., 2023), it does not offer a direct performance comparison with these methods. Since both approaches leverage graph representations to model filter importance, comparing them directly could help clarify the advantages or limitations of the proposed method.
5. Lack of Qualitative Insights on Pruned Model Structure: The paper lacks qualitative insights into the structural impact of pruning. For instance, it would be valuable to see if the graph-based pruning method preserves specific types of filters, or if it tends to prune filters that correspond to certain types of visual features.

**Questions:**

I have the following questions, subject to satisfactory responses, I am open to increasing the score.

1. Could you clarify the computational complexity introduced by the graph construction, random walk, and eigenvalue analysis? Specifically, how does the method scale with network size and dataset complexity?
2. How sensitive is the pruning method to changes in key hyperparameters, such as scaling or the pruning strength parameter. Did you conduct experiments to tune these for each model and dataset?
3. How well does the proposed graph-based pruning approach generalize to other CNN architectures? It seems from the experiments that larger architectures result in more improvements?
4. Have you considered comparing this approach directly with other graph-based pruning techniques, such as those leveraging Von Neumann graph entropy (e.g., Shi et al., 2023)? If yes, where is the result?
5. Did you test the pruned models in scenarios that mimic real-world deployment settings, such as inference on edge devices or in latency-constrained environments? How do the pruned models perform in terms of inference speed or memory footprint in such contexts?
6. How sensitive is the Layer Importance Score (LIS), based on the largest eigenvalue, to variations in network depth and width? Does this eigenvalue-based metric scale consistently across larger or smaller networks?

Minor Issues and Typos

1. Page 2:  “some filters can have similar functionality, our approach assumes…” — This sentence is a run-on and could be revised to “some filters can have similar functionality. Our approach assumes…”
2. Equation Labels**: Some equations, such as those on page 5 for transition probabilities, would benefit from explicit numbering for easier reference within the text.

---

### Official Review · Reviewer_UqAQ · 2024-11-03

**Soundness:** 3
**Presentation:** 2
**Contribution:** 2
**Rating:** 3
**Confidence:** 2

**Summary:**

This paper introduces a filter pruning method by leveraging a graph-based model to capture the global relationships among filters in convolutional layers. By representing each filter as a node in a directed graph and modeling the linear dependencies between filters through edges, the method assesses the relative importance of each filter via random walks on the graph. Experiments on CIFAR-10 and ImageNet demonstrate the superiority of the proposed method over state-of-the-art techniques.

**Strengths:**

1. The paper presents an interesting idea by considering the global relations among filters for network pruning, moving beyond traditional pairwise similarity measures.

2. The authors conduct sufficient experiments on standard datasets like CIFAR-10 and ImageNet, demonstrating the effectiveness of their method compared to existing approaches.

**Weaknesses:**

1. Unclear motivation in introduction:
- The third paragraph of the introduction appears redundant. If the authors aim to provide background information, the detailed descriptions of other methods might be better placed in the related work section. Alternatively, if the intention is to highlight the limitations of existing graph-driven methods to underscore the importance of their work, a more in-depth analysis is needed rather than concluding with a brief statement like "however, they only focus on the approximated error and haven’t dived into the relationships among filters."

- The fourth paragraph of the introduction uses a single sentence to point out the work's motivation: "However, most of these approaches predominantly rely on similarity metrics like Euclidean distance or cosine similarity to measure the similarity between filters, which only considers the pairwise relationships between filters, overlooking the interactions among them within the whole layer." This is insufficient to convey the depth of the motivation. The authors should expand on why pairwise operations overlook the underlying global information and how this impacts the effectiveness of pruning methods.

2. Lack of discussion on transformers: The paper lacks discussion on transformers, which are inherently dense deep neural network modules with significant pruning potential. Considering the conceptual connections between transformers and graph neural networks, it would be valuable to explore whether the proposed graph-driven filter pruning method could be even more effective on transformer architectures.

3. Insufficient theoretical justification: There is a lack of theoretical justification to demonstrate the advantages of the proposed method.

4. Outdated related works and missing references: The related work section is somewhat outdated and misses important recent studies [1, 2, 3].

5. Lack of discussion regarding the potential limitations of the proposed method.


[1] Interpretable Task-inspired Adaptive Filter Pruning for Neural Networks Under Multiple Constraints

[2] Pruning from Scratch via Shared Pruning Module and Nuclear norm-based
Regularization

[3] Structure-Preserving Network Compression Via Low-Rank Induced Training Through Linear Layers Composition

**Questions:**

See weaknesses.

---

### Official Review · Reviewer_wTJ1 · 2024-11-04

**Soundness:** 3
**Presentation:** 3
**Contribution:** 2
**Rating:** 5
**Confidence:** 4

**Summary:**

This paper presents a novel approach to filter pruning in convolutional neural networks using graph theory and random matrix analysis.

**Strengths:**

### Novel Theoretical Framework:

1.Innovative combination of graph theory and random matrix theory.

2. Leverages PageRank concepts for filter importance evaluation.

3. Provides global perspective on filter relationships versus traditional pairwise measures.


### Mathematical Foundation:

1.Links filter pruning to well-established random matrix theory.

2. Uses eigenvalue spectrum analysis for layer importance.

3. Provides systematic approach to identify redundant filters.


### Global Information Capture:

1. Moves beyond simple pairwise relationships.

2. Considers network-wide dependencies.

3. More comprehensive than traditional distance-based methods.

**Weaknesses:**

### Methodological Gaps:

1. Edge weight construction process is not clearly defined.

2.Unclear choice between potential similarity metrics (Euclidean, cosine, inner product).

3.Lack of justification for specific graph construction choices.


### Limited Evaluation:
1. Comparisons mainly with older methods.

2. Limited benchmark datasets.

3. Missing comparisons with recent state-of-the-art pruning methods.


### Theoretical Integration:

1. Random matrix theory and graph-based approach treated separately despite potential unification.

2. Missing theoretical connection between these two perspectives.

3. Incomplete justification of random matrix assumptions.

**Questions:**

### Methodology Clarity:

1. Clearly define edge weight calculation method.

2. Provide ablation studies on different similarity metrics.

3. Explain the rationale behind graph construction choices.

### Theoretical Development:

1. Unify graph-based and random matrix theory perspectives.

2. Provide theoretical bounds on pruning performance.

3. Analyze the relationship between eigenvalue distribution and network performance.

---

### Official Review · Reviewer_SXB4 · 2024-11-04

**Soundness:** 2
**Presentation:** 2
**Contribution:** 2
**Rating:** 3
**Confidence:** 3

**Summary:**

This paper proposes a novel filter-pruning method for resource-constrained environments. This method effectively quantifies structural redundancy, reducing storage and computational costs while maintaining high model accuracy. Experiments show that this method has a certain level of effectiveness.

**Strengths:**

1. The motivation behind this approach is clear, and researching the filter pruning method for resource-constrained environments is meaningful.
2. The authors provide a detailed description of the proposed method, making it easy to follow and reproduce.

**Weaknesses:**

1. This paper only conducts experiments on the CIFAR10 and ImageNet datasets, lacking testing on larger-scale datasets. Additionally, the comparison methods are relatively outdated, missing comparisons with the latest SOTA methods.
2. The study only explores the effectiveness on CNNs. Would this method also be effective on Transformer models?
3. The paper lacks ablation studies and does not include sensitivity analysis on key parameters.
4. There is no analysis of the algorithm's time complexity. At the same time, the results indicate that the performance improvement is modest, raising the question of whether the additional time cost is justified.
5. Why does Table 3 only show reduction rates for Params and FLOPs, instead of specific values as shown in Table 2?
6. The proposed method lacks theoretical support.
7. The paper contains some writing errors, such as a repeated period in line 516. Additionally, the text in the images is too small, making it difficult to read.

**Questions:**

Please see Weaknesses.

---

### Note · Authors · 2024-11-22

I have read and agree with the venue's withdrawal policy on behalf of myself and my co-authors.